# MOVINGPARTS: MOTION-BASED 3D PART DISCOVERY IN DYNAMIC RADIANCE FIELD

**Kaizhi Yang[1], Xiaoshuai Zhang[2], Zhiao Huang[2], Xuejin Chen[1], Zexiang Xu[3,*], Hao Su[2,†,*]**

[1] University of Science and Technology of China, [2] University of California, San Diego,
[3] Adobe Research

ykz0923@mail.ustc.edu.cn, {xiz040, z2huang, haosu}@eng.ucsd.edu,
xjchen99@ustc.edu.cn, zexu@adobe.com

## ABSTRACT

We present MovingParts, a NeRF-based method for dynamic scene reconstruction and part discovery. We consider motion as an important cue for identifying parts, that all particles on the same part share the common motion pattern. From the perspective of fluid simulation, existing deformation-based methods for dynamic NeRF can be seen as parameterizing the scene motion under the Eulerian view, i.e., focusing on specific locations in space through which the fluid flows as time passes. However, it is intractable to extract the motion of constituting objects or parts using the Eulerian view representation. In this work, we introduce the dual Lagrangian view and enforce representations under the Eulerian/Lagrangian views to be cycle-consistent. Under the Lagrangian view, we parameterize the scene motion by tracking the trajectory of particles on objects. The Lagrangian view makes it convenient to discover parts by factorizing the scene motion as a composition of part-level rigid motions. Experimentally, our method can achieve fast and high-quality dynamic scene reconstruction from even a single moving camera, and the induced part-based representation allows direct applications of part tracking, animation, 3D scene editing, etc.

## 1 INTRODUCTION

3D scene reconstruction and understanding is one of the central problems in computer vision and graphics, with a wide range of applications in mixed reality, robotics, movie production, etc. While many works focus on static scenes, real-world physical scenes are usually dynamic and entangled with illumination changes, object motion, and shape deformation. The reconstruction of dynamic scenes is known to be highly challenging. Nonrigid structure from motion methods (Bregler et al. (2000); Gotardo & Martinez (2011); Kong & Lucey (2019); Sidhu et al. (2020)) could recover nonrigid shapes but are limited to sparse feature tracking. To reduce the ambiguity between shape and motion, some other methods introduce multi-view capture (Zhang et al. (2003); Oswald et al. (2014); Tung et al. (2009)) or category-specific priors (Egger et al. (2021); Habermann et al. (2019); Kocabas et al. (2020)). Recently, neural radiance representa-

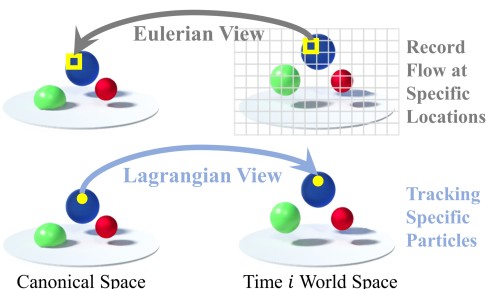

Figure 1: Location-based *Eulerian view* vs. particle-based *Lagrangian view*. The Eulerian view observes the flow at a specific location (akin to the approach taken by deformation-based dynamic NeRF) and the Lagrangian view observes the trajectory of specific particles. These two views constitute the conversion between the dynamic world space of each temporal frame and a static canonical space.

tion NeRF (Mildenhall et al. (2020)) has been applied to this field and achieved promising dynamic capture performance on general scenes using only monocular input (Pumarola et al. (2020); Park et al. (2021a); Li et al. (2022)). However, most NeRF-based dynamic scene modeling methods only focus on scene reconstruction without considering scene understanding, thus lacking the ability to directly support downstream applications that need tracking, shape editing, re-animation, etc.

---

*denotes equal advisory and † denotes the corresponding author.

Our goal is to enable practical dynamic scene capture with both high-quality reconstruction and meaningful scene understanding from monocular input. To this end, we propose MovingParts, a novel NeRF-based approach that can achieve not only fast dynamic scene reconstruction but also automatic rigid part discovery. Our key insight is that motion (while complicating the reconstruction) is an effective cue for identifying object parts because the scene content belonging to one rigid part has to share the same rigid transformation. Therefore, we design novel modules to explain the motions in dynamic neural fields, enabling unsupervised object part discovery via motion grouping. Since the rigid motion patterns are used as the evidence of part discovery, we make explicit the assumption of our input here, which is the general scene with piece-wise rigid motion.

Our approach is inspired by the literature on fluid simulation. We note that a family of previous dynamic NeRF methods model motion using a 3D field that encodes scene flow (Li et al. (2021)) or deformation (Pumarola et al. (2020); Park et al. (2021a)). Specifically, at time $t$, for each location $\mathbf{x}$, the 3D field encodes which particle $\mathbf{x}_c$ in the canonical frame has been deformed to $\mathbf{x}$, which actually backward deforms the particles from world space to static canonical space. As shown in Figure 1, this is essentially the Eulerian view in fluid simulation (Fedkiw et al. (2001)) – motion information is denoted as a function $\Psi_E(\mathbf{x}, t)$ at each specific location $\mathbf{x}$ in the world coordinate frame. It is known that, while the entire scene motion can be modeled under the Eulerian view, specific object/part motion at different temporal moments is actually intractable and hard to analyze. On the other hand, the Lagrangian view (Macklin et al. (2014)), as the duality of the Eulerian view, uses the particle-based representation to track the motion of each particle belonging to the object, which forward deforms the particles from canonical space to world space. The constructed particle trajectory from the Lagrangian view can be an important clue for scene analysis. A relevant Lagrangian-based work is Watch-It-Move (Noguchi et al. (2022)) which composes objects into several ellipsoid-like parts by rendering supervision. However, the multi-view requirement and the ellipsoidal geometric prior highly limit its application. In contrast, we mainly focus on monocular input.

To achieve meaningful scene understanding by motion analysis, we propose a hybrid approach that learns motion under both the Eulerian and the Lagrangian views. In particular, our neural dynamic scene model consists of three modules: (1) a canonical module that models the scene geometry and appearance as a radiance field in a static canonical space, (2) an Eulerian module $\Psi_E(\mathbf{x}, t)$ that records which particle $\mathbf{x}_c$ in the canonical space passes through each specific location $\mathbf{x}$ in the world coordinate frame at every time step, and (3) a Lagrangian module $\Psi_L(\mathbf{x}_c, t)$ that records the trajectory of all particles $\mathbf{x}_c$ in the canonical space. Note that the motions modeled by the Eulerian and Lagrangian modules are inherently reciprocal, we, therefore, apply a cycle-consistency loss during reconstruction to enforce the consistency between the two modules, constraining them to model the same underlying motion in the scene.

The construction of the Lagrangian view makes it convenient to discover parts by factorizing $\Psi_L(\mathbf{x}_c, t)$. As the particles in a rigid part share a common rigid transformation pattern, we propose a novel motion grouping module as part of our Lagrangian module. By projecting the particle motion features into a few groups, we divide the scene into meaningful parts. Once reconstructed, our Lagrangian module could offer part-level representation and allow for direct downstream applications such as part tracking, object control, and scene editing. Since the number of rigid parts generally differs across scenes, we introduce an additional post-processing merging module that can adaptively merge the over-segmented groups into a reasonable number of rigid parts.

We jointly train all modules with only rendering supervision. We demonstrate that our approach achieves high-quality dynamic scene reconstruction and realistic rendering results on par with state-of-the-art methods. More importantly, compared with previous monocular NeRF methods, ours is the only one that simultaneously achieves part discovery, allowing for many more downstream applications. Finally, inspired by recent fast NeRF reconstruction methods (Sun et al. (2022); Chen et al. (2022); Yu et al. (2022)), we construct our system with feature volumes and light-weight multi-layer perceptrons (MLPs), leading to a fast reconstruction speed comparable to other concurrent methods that are specifically focused on speeding up dynamic NeRF.

In summary, our key contributions are:
- We propose a novel NeRF-based method for simultaneous dynamic scene reconstruction and rigid part discovery from monocular image sequences;
- The hybrid representation of feature volume and neural network allows us to achieve both high-quality reconstruction and reasonable part discovery within 30 minutes;

- The extracted part-level representation can be directly applied to downstream applications like part tracking, object control, scene editing, etc.

## 2 RELATED WORK

**Dynamic Neural Radiance Fields.** Recently, the emergence of Neural Radiance Fields (NeRF) (Mildenhall et al. (2020)) has facilitated the tasks of scene reconstruction and image synthesis. Due to the dynamic properties of the physical world, an important branch of NeRF research is to extend it to dynamic scenes (Pumarola et al. (2020); Li et al. (2021); Park et al. (2021b); Fridovich-Keil et al. (2023); Cao & Johnson (2023)). In particular, some methods directly extend the 5D radiance field function to 6D by adding additional time-dependent input to the network (Li et al. (2022); Xian et al. (2021)). Other works enhance the temporal consistency in the 6D dynamic radiance field by explicitly modeling dynamic scene flows (Li et al. (2021); Du et al. (2021); Gao et al. (2021)), leading to promising results from only monocular input. Meanwhile, deformation modules have also been adopted in NeRF-based methods (Pumarola et al. (2020); Tretschk et al. (2021); Yuan et al. (2021); Park et al. (2021a;b); Liu et al. (2022)), offering strong regularization for temporal consistency. Note that these various NeRF-based methods all explain motions (modeled as flows or deformation fields) from the location-based Eulerian view and do not support part discovery. We instead propose a hybrid model that models motions with both location-based Eulerian and particle-based Lagrangian views, enabling high-quality dynamic scene reconstruction with automatic part discovery based on particle motion. In addition to these general methods, some NeRF methods have been devised for particular domains, such as humans (Jiakai et al. (2021); Noguchi et al. (2021); Weng et al. (2022); Peng et al. (2023)), and articulated objects within specific categories Wei et al. (2022). While capable of achieving dynamic scene rendering and part segmentation, these methods often incorporate category priors into the pipeline and cannot be directly applied to general objects.

**Part Discovery from Motion.** At the image level, most motion-based object discovery methods (Keuper et al. (2015); Pia Bideau (2016); Yang et al. (2021); Xie et al. (2019); Papazoglou & Ferrari (2013)) employ the clustering of 2D pixels based on features related to optical flow. We share a common underlying logic with these 2D methods that discover parts (or objects) by constructing and grouping motion trajectories. However, in contrast to these approaches, our method establishes a motion group module on canonical 3D particles and relies on predicted 3D rigid motion, which ensures arbitrary viewpoints consistency and temporal consistency of the grouping results. In the 3D domain, some methods (Shi et al. (2021); Kawana et al. (2022)) reason about object parts by constructing point-wise correspondence at different object states and clustering their trajectories. Without 3D input, (Agudo & Moreno-Noguer (2019)) adopts non-rigid structure from motion to reconstruct the 3D shape and applies spatio-temporal clustering to the 3D points to reason about segmentation. However, only the geometry of sparse feature points could be achieved. Recently, NeRF-based dynamic scene decoupling methods (Yuan et al. (2021); Tschernezki et al. (2021); Wu et al. (2022)) have been proposed. Although they achieve dynamic scene decomposition with high-quality reconstruction, they can only divide the scene into static/dynamic parts and are unable to identify motion patterns. A relevant recent work is Watch-It-Move (Noguchi et al. (2022)), which achieves high-quality part-level reconstruction from image sequences. However, it requires dense multi-view input and imposes ellipsoid-like priors to the part geometry, which may completely fail on challenging monocular data with complex scene geometry. In contrast, our NeRF-based method does not require any shape priors of dynamic objects in complex scenes and can achieve dynamic reconstruction and part discovery from monocular input.

## 3 PRELIMINARIES: NERF AND D-NERF

By incorporating implicit function and volume rendering, Neural Radiance Field (NeRF) (Mildenhall et al. (2020)) allows for scene reconstruction and novel view synthesis via optimizing scene representation directly. In general, NeRF interprets static scenario as a continuous implicit function $F_\theta$. By querying spatial coordinates ($\mathbf{x}$) and view direction ($\mathbf{d}$), $F_\theta$ outputs the corresponding density ($\sigma$) and observed color ($\mathbf{c}$) as $(\mathbf{c}, \sigma) = F_\theta(\mathbf{x}, \mathbf{d})$. Through classical volume rendering in graphics, the 3D scene representation $F_\theta$ can be rendered into a 2D image. Specifically, given a ray $\mathbf{r}$ emitted from the optical center to a specific pixel in the image, the rendered color of that pixel is an integral of all the colors on the ray with near and far bounds $h_n$ and $h_f$:

$$\mathbf{C}(\mathbf{r}) = \int_{h_n}^{h_f} T(h)\sigma\big(\mathbf{r}(h)\big)\mathbf{c}\big(\mathbf{r}(h), \mathbf{d}\big)dh, \quad \text{where} \quad T(h) = \exp\Big(-\int_{h_n}^{h} \sigma(\mathbf{r}(s))ds\Big). \quad (1)$$

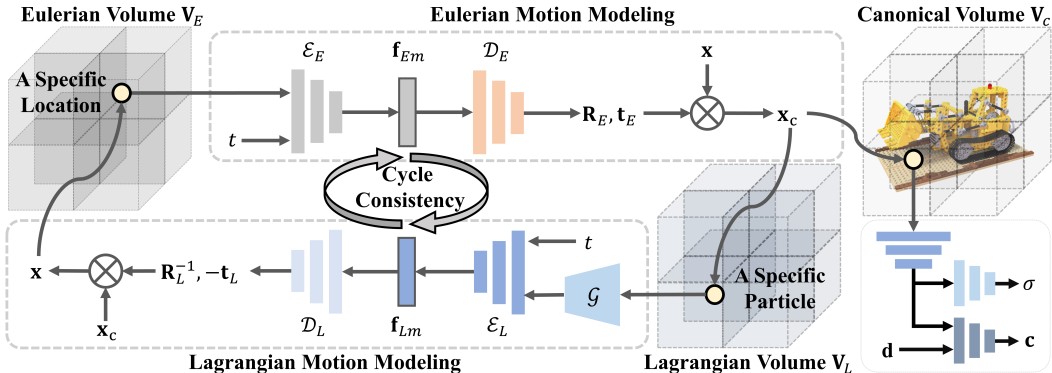

Figure 2: Overview of our method. Inspired by the Eulerian and Lagrangian viewpoints in fluid simulation, we designed three modules for motion-based part discovery in a scene. The Eulerian module and the Lagrangian module observe the motion of specific spatial locations and specific particles, respectively. They both comprise a mutual mapping of a point between its position at an arbitrary time instance and its canonical configuration. The canonical module serves to reconstruct the geometry and appearance for volume rendering. Based on the particle trajectories recorded by the Lagrangian module, we can analyze the motion patterns and discover rigid parts.

$T(h)$ can be interpreted as the transparency accumulated from $h_n$ to $h$. Because of the inherent differentiability of Eq. 1, it only requires a set of images with camera poses to optimize $F_\theta$ directly.

D-NeRF (Pumarola et al. (2020)) extends NeRF to capture dynamic scenes, assuming that there is a static canonical space and includes all objects. It divides the dynamic scene reconstruction in world space into two sub-problems: the NeRF representation learning of canonical space and the learning of the mapping from the world space to the canonical space (scene flow prediction) as:

$$(\mathbf{c}, \sigma) = F_\theta(\Psi(\mathbf{x}, t), \mathbf{d}) \tag{2}$$

where $\Psi(\mathbf{x}, t)$ predict the canonical space position from $\mathbf{x}$ at time $t$ into its canonical configuration.

## 4 OUR METHOD

From the perspective of fluid simulation, the scene motion is composed of particle motions. $\Psi(\mathbf{x}, t)$ in D-NeRF can be interpreted as recording the motion of particles passing through a given coordinate $\mathbf{x}$ at time $t$, corresponding to the Eulerian perspective. Following D-NeRF, we also assume a canonical space that is static and includes all objects. Besides the Eulerian perspective, we also describe the dynamic scene from the Lagrangian perspective. Accordingly, we construct three modules, as Figure 2 shows, that include an Eulerian module $\Psi_E(\mathbf{x}, t)$ which maps a position $\mathbf{x}$ at any time $t$ in the world space to the canonical space, a Lagrangian module $\Psi_L(\mathbf{x}_c, t)$ which tracks the trajectory of a particle corresponding to $\mathbf{x}_c$ in the canonical space, and a canonical module which encodes the appearance and geometry in the canonical scene. Under the assumption of finite rigid bodies, we exploit the learned motion by the Lagrangian module and design a motion grouping module to discover moving parts. The particles in the same group share a common rigid transformation and should belong to the same part. Next, we will describe these modules and loss functions in detail.

### 4.1 CANONICAL MODULE

Same as NeRF, the canonical module is formulated as an implicit function $F_\theta(\mathbf{x}_c, \mathbf{d}) \rightarrow (\mathbf{c}, \sigma)$ which encodes the geometry and appearance in a canonical space. To accelerate convergence, a hybrid representation of feature volume and neural network is adopted. The queried canonical coordinate $\mathbf{x}_c$ is first used to interpolate the corresponding features within a 3D feature volume $\mathbf{V}_c \in \mathbb{R}^{N_x \times N_y \times N_z \times C}$, where the $N_x \times N_y \times N_z$ denotes the spatial resolution and $C$ is the feature dimension. To alleviate the local gradient artifact of grid representation, we adopt multi-distance interpolation and concatenate the features in different resolutions as (Fang et al. (2022)):

$$\mathbf{f}_c = \text{Tri-Interp}(\mathbf{x}_c, \mathbf{V}) \oplus ... \oplus \text{Tri-Interp}(\mathbf{x}_c, \mathbf{V}[:: s_M]). \tag{3}$$

After positional encoding, the queried feature with $\mathbf{d}$ is fed into MLPs to predict $\sigma$ and $\mathbf{c}$.

## 4.2 EULERIAN MODULE

The Eulerian module $\Psi_E(\mathbf{x}, t)$ records which particle $\mathbf{x}_c$ in the canonical space goes through a specific location $\mathbf{x}$ at the query time $t$. Assuming that the scene is piece-wise rigid, we formulate this mapping as a rigid transformation in $\mathbb{SE}(3)$ similar to Park et al. (2021a), which ensures that all the points on the same rigid body can be transformed using the same set of parameters. Specifically, our Eulerian module contains three components. First, the 3D feature volume $\mathbf{V}_E$ stores the information about the particles that pass through each position during the entire observation period. Second, a motion extractor $\mathcal{E}_E$ decodes the motion feature from the interpolated feature in $\mathbf{V}_E$ at query time $t$. Third, different from Park et al. (2021a) that uses a screw axis as an intermediate representation, our rigid transformation decoder $\mathcal{D}_E$ directly maps the motion feature to rotation and translation parameters. The overall process can be formulated as:

$$(\mathbf{R}_E, \mathbf{t}_E) = \mathcal{D}_E(\mathbf{f}_{Em}), \quad \text{where} \quad \mathbf{f}_{Em} = \mathcal{E}_E\left(\text{Tri-Interp}\left(\mathbf{x}, \mathbf{V}_E\right), t\right) \tag{4}$$

Here we employ the continuous 6D intermediate representation (Zhou et al. (2019)) for 3D rotation $\mathbf{R}_E$. The Eulerian mapping from the world space at each temporal frame to the canonical space can be calculated by:

$$\mathbf{x}_c = \mathbf{R}_E(\mathbf{x} - \mathbf{t}_E) \tag{5}$$

## 4.3 LAGRANGIAN MODULE

As the inverse of the Eulerian module, the Lagrangian module $\Psi_L(\mathbf{x}_c, t)$ tracks the trajectories of specific object particles over time. We use the same manner to construct $\mathbf{V}_L$, $\mathcal{E}_L$ and $\mathcal{D}_L$. Different from the Eulerian perspective, the trajectories of each particle in the Lagrangian perspective can be an important cue for rigid part discovery. All particles belonging to the same rigid part share the same rigid body transformation, which means that their motion can be represented by a single feature vector. So we add an additional motion grouping network $\mathcal{G}$ (see Figure 3) after $\mathbf{V}_L$ to restrict that particle trajectories are only subject to a finite number of rigid motion patterns.

Similar to (Xu et al. (2022)), we use the attention module with the straight-through estimator trick to achieve the hard grouping of La-

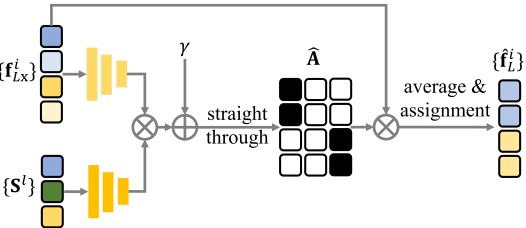

Figure 3: The architecture of our motion grouping network $\mathcal{G}$. We first calculate the similarity between each fused feature $\mathbf{f}_{L\mathbf{x}}^i$ and each learnable slot $\mathbf{S}^l$, and then apply Gumbel-softmax with the straight-through trick to achieve hard grouping $\hat{\mathbf{A}}$. Finally, we assign $\hat{\mathbf{f}}_L^i$ to be the average of $\{\mathbf{f}_L^i\}$ in its corresponding group by Equation 8.

grangian features. To encourage the spatial coherence of points in the same group, the coordinate of each point $\mathbf{x}_c^i$ is concatenated to the corresponding Lagrangian feature $\mathbf{f}_L^i$. Specifically, we first compute the similarity map $\mathbf{A}$ between the feature $\{\mathbf{f}_{L\mathbf{x}}^i = \mathbf{f}_L^i \oplus \mathbf{x}_c^i\}$ and learnable slots $\{\mathbf{S}^l\}$ by Gumbel-softmax:

$$\mathbf{A}_{il} = \frac{\exp(\mathbf{W}_q \mathbf{f}_{L\mathbf{x}}^i \cdot \mathbf{W}_k \mathbf{S}^l + \gamma_i)}{\sum_1^L \exp(\mathbf{W}_q \mathbf{f}_{L\mathbf{x}}^i \cdot \mathbf{W}_k \mathbf{S}^l + \gamma_i)}, \tag{6}$$

where $\mathbf{W}_q$ and $\mathbf{W}_k$ are linear mappings and $\gamma$ is a sample drawn from $Gumbel(0, 1)$. Then the straight-through estimator trick is used to convert the soft similarity map to one-hot formulation:

$$\hat{\mathbf{A}} = \text{one-hot}(\mathbf{A}_{\text{argmax}}) + \mathbf{A} - \text{detach}(\mathbf{A}) \tag{7}$$

where the detach operation cuts off the corresponding gradient. Despite the hard conversion, Equation 7 can still keep the gradient the same as $\mathbf{A}$. The hard similarity matrix $\hat{\mathbf{A}}$ distributes all the Lagrangian features into several groups, where each group represents the particles with the same motion pattern. Instead of directly assigning the learned slot as the updated Lagrangian feature, we calculate the average of all the Lagrangian features in the same group to update the original Lagrangian features. In this way, each updated Lagrangian feature will be directly related to the Lagrangian grid $\mathbf{V}_L$, allowing for more efficient optimization. This procedure can be formulated as:

$$\hat{\mathbf{f}}_L^i = \frac{\sum_1^I \hat{\mathbf{A}}_{il} \cdot \mathbf{f}_L^i}{\sum_1^I \hat{\mathbf{A}}_{il}} \tag{8}$$

Then the updated Lagrangian features $\hat{\mathbf{f}}_L^i$ with query time $t$ is fed into $\mathcal{E}_L$ and $\mathcal{D}_L$ sequentially to decode the motion feature $\mathbf{f}_{Lm}$ and the rigid transformation $\mathbf{R}_L, \mathbf{t}_L$. As mentioned in Section 4.4, to efficiently implement the cycle consistency between the Eulerian and Lagrangian modules, we expect $\mathbf{R}_L = \mathbf{R}_E$ and $\mathbf{t}_L = \mathbf{t}_E$. So the Lagrangian mapping from the canonical space to the world space at each temporal frame is calculated by:

$$\mathbf{x} = \mathbf{R}_L^{-1}(\mathbf{x}_c + \mathbf{t}_L) \tag{9}$$

## 4.4 Loss Functions

As our main optimization goal, we adopt the Mean Squared Error (MSE) between the rendered pixel color and the ground truth pixel color as our reconstruction loss:

$$\mathcal{L}_{\text{photo}} = \frac{1}{|\mathcal{R}|} \sum_{\mathbf{r} \in \mathcal{R}} \|\hat{\mathbf{C}}(\mathbf{r}) - \mathbf{C}(\mathbf{r})\|_2^2. \tag{10}$$

We also use a total variation loss $\mathcal{L}_{\text{tv}}$ to smooth the motion volumes and encourage motion similarity of spatial neighbors. Following Sun et al. (2022), the per-point color loss $\mathcal{L}_{\text{per\_pt}}$ and background entropy loss $\mathcal{L}_{\text{entropy}}$ are used to directly supervise the sampled point color and encourage the background probability to concentrate around 0 or 1.

In addition, a cyclic consistency loss is designed to encourage the reciprocity of the Lagrangian module and the Eulerian module. Instead of measuring the displacement of the transformations between these two views like Liu et al. (2022), we found that accounting for the difference between low-level motion features $\mathbf{f}_{Lm}$ and $\mathbf{f}_{Em}$ leads to more robust optimization and better part discovery. Our cycle loss is defined as:

$$\mathcal{L}_{\text{cycle}} = \frac{1}{|\mathcal{P}_{obj}|} \sum_{\mathbf{x} \in \mathcal{P}_{obj}} \|\mathbf{f}_{Lm}^{\mathbf{x}} - \mathbf{f}_{Em}^{\mathbf{x}}\|_2^2. \tag{11}$$

Please refer to Appendix A.1 for a more detailed discussion of these two implementations of the cyclic consistency loss. Since the deformation of free space does not satisfy the assumption of finite rigid motions, we filter out free space according to density value and only calculate $\mathcal{L}_{\text{cycle}}$ at sampled points on objects $\{\mathbf{x} \in \mathcal{P}_{obj} | \sigma_{\mathbf{x}} > \epsilon\}$. In our experiments, $\epsilon = 10^{-4}$. The overall loss function is:

$$\mathcal{L} = \mathcal{L}_{\text{photo}} + w_{\text{cycle}}\mathcal{L}_{\text{cycle}} + w_{\text{per\_pt}}\mathcal{L}_{\text{per\_pt}} + w_{\text{entropy}}\mathcal{L}_{\text{entropy}} + w_{\text{tv}}\mathcal{L}_{\text{tv}}. \tag{12}$$

## 4.5 Group Merging Module

It is not reasonable to use the same number of groups for a variety of scenarios. We generally set a large number of groups as an upper bound on the number of rigid bodies in the scene, which may cause over-segmented results (Figure 5). This is because we provide an excessive number of groups, and also the same rigid transformations could be easily represented by very different high-level motion features. To address this problem, we design an efficient heuristic algorithm for group merging based on motion differences. This algorithm is used as post-processing after training only and does not affect the parameters of the model. We summa-

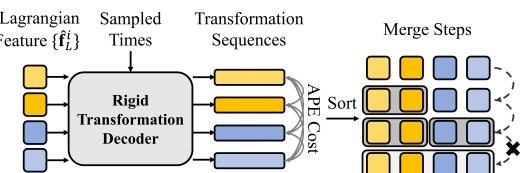

Figure 4: The group merging procedure. We decode the average Lagrangian features of each group into rigid transformation sequences and determine the merge order as well as the stop step by evaluating the APE cost between the sequences.

rize this group merging algorithm in Figure 4. 1) We sample points uniformly in canonical space and filter the free space points with density lower than the threshold $\epsilon$. 2) These remained points are fed into the Lagrangian module to get the updated feature $\hat{\mathbf{f}}_L^i$, which is the high-level representation of each motion group. 3) We evaluate the rigid transformation similarity between each pair of groups: The rigid transformation sequences are generated by decoding the updated slots into rotation and translation with uniformly sampled times between 0 and 1. 4) We use the Absolute Pose Error (APE) to measure the difference between each sequence pair:

$$APE_{i,j} = \sum_t \|(\mathbf{P}_i^t)^{-1}\mathbf{P}_j^t - \mathbf{I}_{4 \times 4}\|, \tag{13}$$

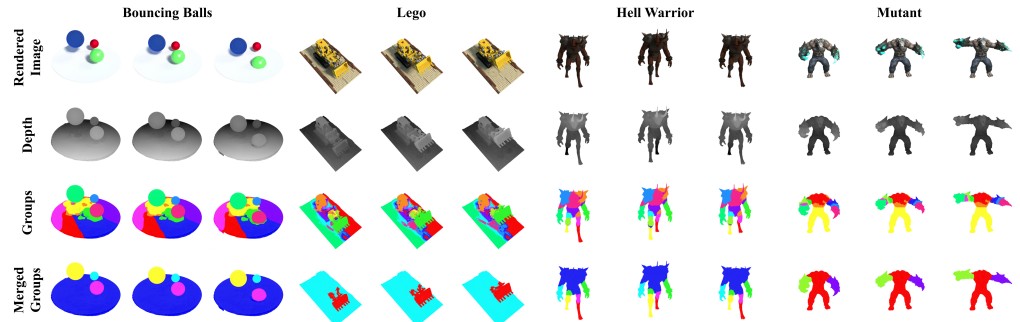

Figure 5: Visualization of our results on the D-NeRF synthetic dataset. The first three rows show our rendered image, depth, and initial grouping result at different timesteps. The last row shows our merged groups which demonstrates the merging module's ability to adaptively aggregate groups with similar motions and achieve clean and accurate segmentation results.

Table 1: Comparison with NeRF-based methods. We indicate the best and second best with bold and underlined markings. We achieve high-quality rendering on par with the state-of-the-art method while reasonably discovering rigid parts, efficiently completing both tasks in about 26 minutes.

| Method | Deformation | Part | Training Time | PSNR ↑ | SSIM ↑ | LPIPS ↓ |
|---|---|---|---|---|---|---|
| T-NeRF | ✗ | ✗ | $\sim$ hours | 29.50 | 0.95 | 0.08 |
| K-Planes | ✗ | ✗ | 52 mins | 31.61 | **0.97** | - |
| HexPlane | ✗ | ✗ | 10 mins | 31.04 | **0.97** | **0.04** |
| D-NeRF | Eulerian | ✗ | 20 hours | 30.43 | 0.95 | 0.07 |
| NDVG | Eulerian | ✗ | 23 mins | 30.54 | 0.96 | 0.05 |
| TiNeuVox | Eulerian | ✗ | 28 mins | **32.67** | **0.97** | **0.04** |
| WIM | Lagrangian | ✓ | $\sim$ hours | 15.72 | 0.83 | 0.19 |
| Ours | Eulerian&Lagrangian | ✓ | 26 mins | 32.18 | **0.97** | **0.04** |

where $\mathbf{P}_i^t$ is the transformation matrix of group $i$ at time $t$. 5) We recursively find the two groups with the smallest APE at the current step and record their merge APE cost until all the groups are merged into a single one. In the early stages, the groups with similar motion patterns are merged, which keeps the merging cost growth slow. Once groups representing different motions are merged, the cost will jump, indicating that the merging process should terminate. In practice, we simply find the termination step with the largest cost increase to the subsequent step as our final result.

## 5 EXPERIMENTS AND RESULTS

Our method not only enables high-quality dynamic scene reconstruction but also allows for the discovery of reasonable rigid parts. In this section, we first evaluate the reconstruction and part discovery performance of our method on the D-NeRF 360° synthetic dataset. Then, we construct a synthetic dataset with ground-truth motion masks to quantitatively evaluate our motion grouping results. Finally, we provide direct applications for structural scene modeling and editing.

### 5.1 IMPLEMENTATION

We use $50 \times 50 \times 50$ voxels for the Eulerian and Lagrangian volume and a $160 \times 160 \times 160$ voxel for the canonical volume. Following Fang et al. (2022), we employ the progressive upsample the resolution for acceleration. We use two separated linear layers to predict the 6D rotation and 3D translation with biases as $(1, 0, 0, 0, 1, 0)$ and $(0, 0, 0)$, respectively, so that the initial deformation is an identity. We use the Adam optimizer for a total of $20k$ iterations, by sampling 4096 rays from a randomly sampled image in each iteration. All the experiments were conducted on a single NVIDIA RTX3090 GPU. More details can be found in the appendix.

### 5.2 EVALUATION ON D-NERF DATASET

We adopt the 360° Synthetic dataset provided by D-NeRF (Pumarola et al. (2020)) to evaluate our method quantitatively and qualitatively. The dataset contains eight synthetic dynamic scenes with different motion patterns, and only one view is captured at each time step. We compare our method with the state-of-the-art dynamic NeRF methods: Non-deformation-based methods T-NeRF (Pumarola et al. (2020)), K-Planes (Fridovich-Keil et al. (2023)), HexPlane (Cao & Johnson (2023)),

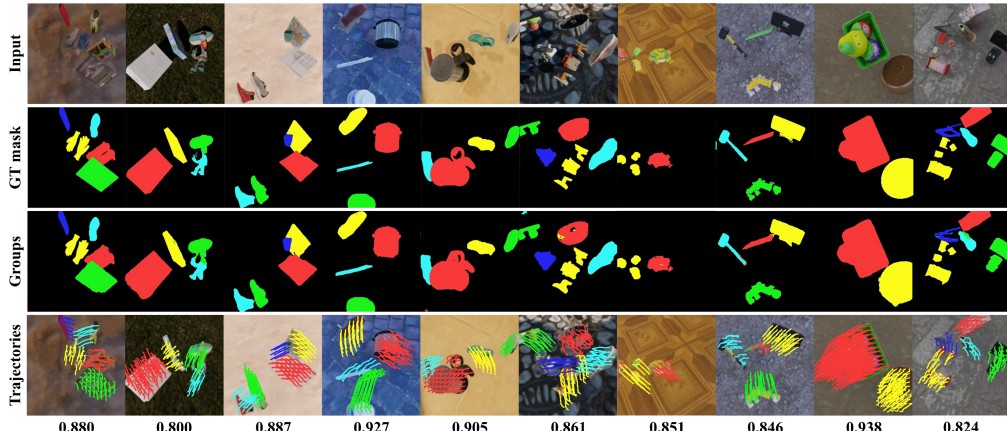

Figure 6: Motion grouping evaluation on our generated datasets. Our method is able to achieve high mIOU scores across various scene configurations and has demonstrated the ability to continuously track specific parts as well as handle complex geometry and topology.

Eulerian-based method D-NeRF (Pumarola et al. (2020)), TiNeuVox (Fang et al. (2022)), NDVG (Guo et al. (2022)) and a Lagrangian-view method WIM (Noguchi et al. (2022)). For TiNeuVox, we use their base version with a canonical grid in $160^3$ resolution and hidden layers of 256 channels. Following these previous works, we train each scene with images at $400 \times 400$ resolution and use two metrics for evaluation: Peak Signal-to-Noise Ratio (PSNR), Structural Similarity (SSIM) (Zhou et al. (2004)) and Learned Perceptual Image Patch Similarity (LPIPS) (Richard et al. (2018)).

As shown in Table 1, while keeping the training time within 30 minutes on one GPU, our method not only achieves high rendering quality but also supports part discovery. Compared to the previous methods, we achieved the best in SSIM and LPIPS, and second best in PSNR. Compared to TiNeu-Vox, our method has a slight PSNR drop. The main reason is that TiNeuVox employs a temporal enhancement module in the canonical space to improve quality, which also leads to a time-varying canonical space. After removing this enhancement module in TiNeuVox, its average PSNR drops to 31.47. In our paper, to achieve better disentanglement of geometry and motion, we expect the geometric evolution only comes from the scene motion. Therefore we did not adopt a similar enhancement strategy to form a time-invariant canonical space. For WIM, due to the nonexistence of canonical space, the significant motion ambiguity under the single view setting causes the failure.

We show our visualization results in Figure 5. It can be seen that our method enables high-quality appearance and geometry reconstruction. We also assign each query point the corresponding group color and render it to 2D images. As discussed in Section 4.5, over-segmentation occurs because similar motion could be represented by different high-level features (see the third row in Figure 5). Through our group merging algorithm, we only retain the highly distinguishable motion modes and obtain concise part segmentation. Thanks to the motion-based grouping mechanism, our method is capable of overlooking motion-irrelevant characteristics in geometry and appearance and producing clean part discovery results on these realistic complex scenes.

## 5.3 MOTION GROUPING EVALUATION

In this section, we provide a quantitative evaluation of our motion grouping results. We created a synthetic dataset with ground truth image-segmentation pairs using Kubric toolkit (Greff et al. (2022)). Each created scene contains 1 to 5 realistic real-world objects from the GSO dataset (Downs et al. (2022)) with different initial velocities and motion directions. We followed the same sampling and rendering process as D-NeRF (Pumarola et al. (2020)) to generate a 120-frame monocular image sequence with $256 \times 256$ resolution for each scene.

To begin our evaluation, we first establish the correlation pairs between the ground truth label and our predicted groups. For each group, we assign the ground truth label with the highest number of pixels corresponding to it in the first 10 frames. More details are included in the appendix. We calculate the mean Intersection over Union (IOU) for the assigned label mask with its corresponding ground truth mask over the entire image sequence. It is noted that achieving a high mIOU score over the entire sequence requires more than just the ability to accurately distinguish each individual part. It also necessitates the capacity to consistently track each part throughout the sequence.

We present 10 examples in Figure 6, showcasing both quantitative mIOU and qualitative visualization and comparisons. Despite the variation in the scene configurations, our method achieves an mIOU score of over $85\%$ on most scenes, clearly demonstrating its robustness over the dataset. Moreover, the high mIOU score indicates that our method can generate accurate part segmentation results and continuously track specific parts throughout the sequences, see the learned trajectories in Figure 6, ensuring both temporal and multi-view consistency of the discovered parts. Furthermore, our method is capable of dealing with complex geometry and topology. Holes (cable in example 5) and geometry details are nicely revealed by our method. By utilizing our motion-based grouping approach, our method can accurately segment objects even if they are spatially separated– see the gloves (example 1) and 3-car (example 7) in Figure 6.

## 5.4 APPLICATION: STRUCTURED SCENE MODELING BY ROBOTIC MANIPULATION

Observation and interaction are crucial for human beings to learn from the real world. In this section, we show that our method can identify objects and understand the functionality of their parts by observing physical interaction procedures. To demonstrate this, we capture a set of robotic manipulation sequences with a similar monocular camera setting as (Pumarola et al. (2020)). As shown in Figure 7 above, by observing the robot's work process, like picking up a toy or inserting a peg, our method can accurately identify the manipulated object, as well as the links and joints of the robot. Note that since the robotic arms' trajectories are different in the two sequences, the joints discovered by motion are also different. The discovered 3D parts with their Lagrangian motion could provide a strong prior for downstream functionality reasoning and robotic reinforcement learning tasks.

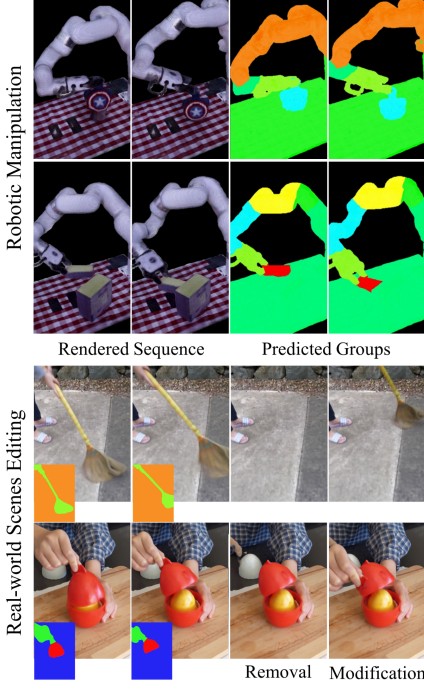

Figure 7: Applications. Our method can directly apply to real-scene structural modeling and editing.

## 5.5 APPLICATION: SCENE EDITING

In addition to scene understanding, with the learned structural representation of dynamic scenes, our method can also edit scenes and generate new renderings from the scene. Figure 7 below presents a few scene-editing applications supported by our approach in HyperNeRF real-world sequence (Park et al. (2021b)). Since our method conducts grouping in the 3D canonical space, the consistency can be maintained not only across multiple views but also across time steps. We show the removal or modification of specific objects in these two real scenes and demonstrate the scalability of our method to real-world applications.

## 6 CONCLUSION

In this paper, we present MovingParts, a novel method for 3D dynamic scene reconstruction and part discovery. Inspired by fluid simulation, we observe the motion in the scene from both the Eulerian view and the Lagrangian view. In the particle-based Lagrangian view, we constrain the motion pattern of the particles to be a few rigid transformations, so that we successfully perform part discovery. To ensure fast convergence during training, we utilize a hybrid feature volume and neural network representation, for both views which are efficiently supervised by a cycle-consistency loss. What is more, the learned part representation could directly be applied to downstream tasks, e.g., object tracking, structured scene modeling, editing, etc.

**Limitations.** Motion modeling at a specific location can be considered as a sequence decoding task. In this paper, we explicitly store the motion features in low-dimensional vectors, which makes it challenging to model motion on very long sequences. Although we can circumvent the issue by manually splitting long sequences into shorter ones, a unified long sequence encoding-decoding scheme will still be a more elegant and efficient solution. We defer the exploration of this challenging setting to future work.

ACKNOWLEDGMENTS

Thanks for the real-scenario robotic data built by Litian Liang (UCSD) and Liuyu Bian (THU). This work was supported by the National Natural Science Foundation of China (NSFC) under Grants 62076230 and the Fundamental Research Funds for the Central Universities under Grant WK3490000008.

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

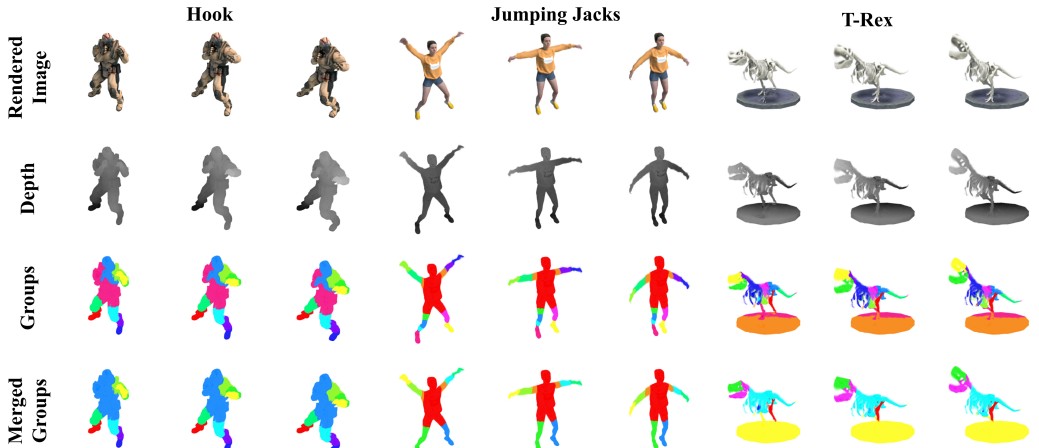

Figure 8: More visualization results of reconstruction and part discovery on D-NeRF synthetic dataset. By analyzing the motion patterns, our method could discover parts that are consistent with common sense, such as human arms and legs.

Table 2: Ablation study about our components and losses, including the Lagrangian module, cycle consistency loss, TV loss, per-point loss, and cross-entropy loss.

|  | $\mathcal{L}_{\text{cycle}}$ | $\mathcal{L}_{\text{tv}}$ | $\mathcal{L}_{\text{entropy}}$ | $\mathcal{L}_{\text{per-pt}}$ | PSNR ↑ | SSIM ↑ | LPIPS ↓ |
|---|---|---|---|---|---|---|---|
| A1 (Ours) | feature-based | ✓ | ✓ | ✓ | 32.183 | 0.971 | 0.036 |
| A2 | ✗ | ✓ | ✓ | ✓ | 32.333 | 0.972 | 0.036 |
| A3 | displacement-based | ✓ | ✓ | ✓ | 31.968 | 0.971 | 0.037 |
| A4 | feature-based | ✗ | ✓ | ✓ | 30.017 | 0.957 | 0.064 |
| A5 | feature-based | ✓ | ✗ | ✓ | 31.988 | 0.971 | 0.037 |
| A6 | feature-based | ✓ | ✓ | ✗ | 32.082 | 0.970 | 0.038 |

# A  APPENDIX

## A.1  ABLATION STUDY

In this section, we conduct ablation experiments on the Lagrangian module and the loss functions to showcase their effectiveness. These ablations were conducted on the D-NeRF synthetic dataset and we report their averaged metric values (PSNR, SSIM and LPIPS) with their corresponding model settings (A1 – 7) in Table 2. Moreover, we show the rendered results of these experiments on the Bouncing Balls scene in Fig. 9 to illustrate the influence of individual modules on rendering and part discovery. We present our full model in Experiment A1.

**Lagrangian module.** Our Lagrangian module is mainly designed for automatic part discovery, which is the main focus of the this work. Without this module, the model cannot achieve part discovery at all while dynamic reconstruction and novel view rendering can still be done. We evaluate how the Lagrangian module is affecting the final rendering with the ablated model setting A2, where we set the weight of the cycle consistency loss to 0, which essentially disables the Lagrangian module. As shown in Table 2, the absence of rigid part motion constraints in the Lagrangian module leads to only a slightly higher PSNR (less than 0.2db difference). In general, our Lagrangian module enables automatic part discovery while retaining high rendering quality.

**Cycle Consistency loss.** To validate the effectiveness of the motion feature-based cycle consistency loss, we compare it with the displacement-based cycle loss in A3. We enforce that the displacement modeled by the Eulerian and Lagrangian modules remains consistent, akin to the Deformation Cycle Consistency in Liu et al. (2022):

$$\mathcal{L}_{\text{cycle-disp}} = \frac{1}{|\mathcal{P}_{obj}|} \sum_{\mathbf{x} \in \mathcal{P}_{obj}} \|\mathbf{x} - \mathbf{R}_L^{-1}(\mathbf{R}_E(\mathbf{x} - \mathbf{t}_E) + \mathbf{t}_L)\|_2^2. \tag{14}$$

As demonstrated in Table 2, there is a decline in PSNR for displacement-based cycle consistency (A3) when compared to the feature-level consistency used in our paper (A1). More importantly, the discovery of parts is contingent upon grouping Lagrangian motion features. The inherent ambiguity

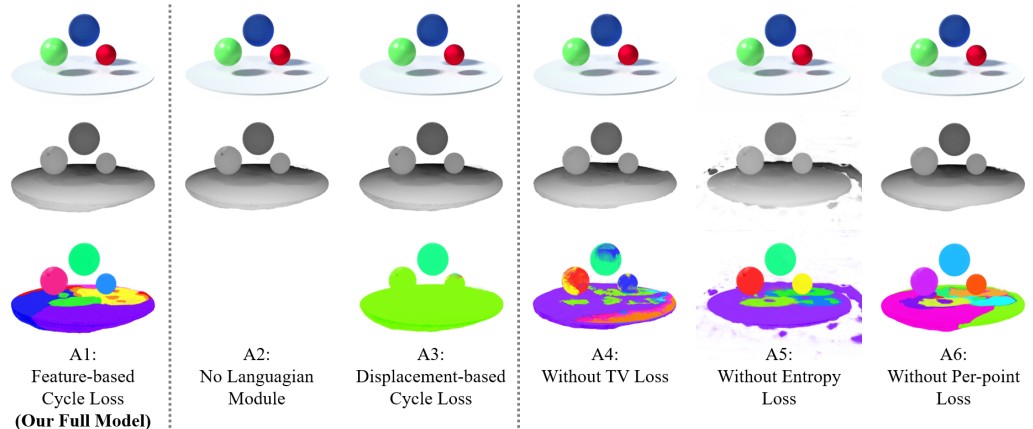

| A1: Feature-based Cycle Loss **(Our Full Model)** | A2: No Languagian Module | A3: Displacement-based Cycle Loss | A4: Without TV Loss | A5: Without Entropy Loss | A6: Without Per-point Loss |

Figure 9: The rendering results of ablations on Bouncing Balls scene.

between motion features and displacement, where multiple motion features may correspond to the same displacement, poses a challenge in attaining precise part discovery results. As a result, the A3 model often fails to distinguish different parts in the scene (see A3 in Figure 9).

**Total variation loss.** The total variation loss imposes constraints on the feature similarity among neighboring grids in the motion volume, effectively ensuring that adjacent particles in space exhibit similar motion patterns. Experiment A4 demonstrates the critical role of this regularization in motion modeling. In its absence, there is a notable decline in image rendering quality. Furthermore, due to the unrestrained movement of near-neighbor particles, the discovered parts lack the characteristic localized nature in 3D space, as illustrated in A4 of Figure 9.

**Additional losses.** The removal of either the cross-entropy loss (A5) or the per-point RGB loss (A6) lead to a decline in rendering quality. Notably, the cross-entropy loss plays a crucial role in regulating foreground-background probability, and its omission may lead to ghosting in the background, as evident in A6 of Figure 9. It is noteworthy that even in the absence of these additional regularization losses, our method can still produce reasonable part discovery results.

## A.2 MORE RECONSTRUCTION AND GROUP RESULTS

In this subsection, we report the per-scene results of the D-NeRF synthetic dataset in Table 3. We also show more dynamic scene reconstruction and part discovery results in Figure 8. It can be seen that our method can achieve high-quality dynamic reconstruction with different motion patterns and also shows the ability that reasonably segment the moving regions and obtain meaningful parts, like the legs and arms. Note that our approach does not explicitly introduce any category and geometric priors. Therefore, it is not necessary to separate parts that are perceptually/semantically distinguishable but have no relative motion with other parts in the video capture, like the head and body.

## A.3 VISUALIZATION OF GROUP MERGING

Due to the excessive number of groups and non-linear neural networks, the same rigid transformation trajectory could be easily represented by different high-level motion features. Therefore, we introduce a group merging module to reduce the number of the group to a plausible level. In this subsection, we visualize two examples of all steps of group merging, as shown in Figure 10 At each merge step, the pair of groups with the current most similar motion patterns are aggregated; the entire process actually builds a binary tree of the original groups. The whole process stops when all groups are combined into a single one, i.e. the step with a single color. Finally, the most reasonable step is selected by

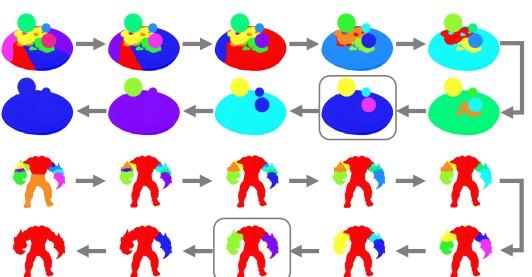

Figure 10: Two examples of the group merging process. The groups with similar motion patterns are gradually merged, and then the most reasonable step could be selected.

Table 3: Per-scene results of D-NeRF synthetic dataset.

| Method | Hell Warrior | | | Mutant | | | Hook | | | Bouncing Balls | | |
|---|---|---|---|---|---|---|---|---|---|---|---|---|
| | PSNR | SSIM | LPIPS | PSNR | SSIM | LPIPS | PSNR | SSIM | LPIPS | PSNR | SSIM | LPIPS |
| T-NeRF | 23.19 | 0.93 | 0.08 | 30.56 | 0.96 | 0.04 | 27.21 | 0.94 | 0.06 | 37.81 | 0.98 | 0.12 |
| D-NeRF | 25.02 | 0.95 | 0.06 | 31.29 | 0.97 | 0.02 | 29.25 | 0.96 | 0.11 | 38.93 | 0.98 | 0.10 |
| TiNeuVox | 28.17 | 0.97 | 0.07 | 33.61 | 0.98 | 0.03 | 31.45 | 0.97 | 0.05 | 40.73 | 0.99 | 0.04 |
| NDVG | 25.53 | 0.95 | 0.07 | 35.53 | 0.99 | 0.01 | 29.80 | 0.96 | 0.04 | 34.58 | 0.97 | 0.11 |
| K-Planes | 25.70 | 0.95 | - | 33.79 | 0.98 | - | 28.50 | 0.95 | - | 41.22 | 0.99 | - |
| HexPlane | 24.24 | 0.94 | 0.07 | 33.79 | 0.98 | 0.03 | 28.71 | 0.96 | 0.05 | 39.69 | 0.99 | 0.03 |
| WIM | 12.35 | 0.81 | 0.21 | 16.20 | 0.85 | 0.16 | 14.16 | 0.82 | 0.19 | 15.82 | 0.84 | 0.29 |
| Ours | 28.66 | 0.97 | 0.04 | 34.42 | 0.98 | 0.02 | 31.39 | 0.97 | 0.04 | 38.99 | 0.99 | 0.04 |
| Method | Lego | | | T-Rex | | | Stand Up | | | Jumping Jacks | | |
| | PSNR | SSIM | LPIPS | PSNR | SSIM | LPIPS | PSNR | SSIM | LPIPS | PSNR | SSIM | LPIPS |
| T-NeRF | 23.82 | 0.90 | 0.15 | 30.19 | 0.96 | 0.13 | 31.24 | 0.97 | 0.02 | 32.01 | 0.97 | 0.03 |
| D-NeRF | 21.64 | 0.83 | 0.16 | 31.75 | 0.97 | 0.03 | 32.79 | 0.98 | 0.02 | 32.80 | 0.98 | 0.03 |
| TiNeuVox | 25.02 | 0.92 | 0.07 | 32.70 | 0.98 | 0.03 | 35.43 | 0.99 | 0.02 | 34.23 | 0.98 | 0.03 |
| NDVG | 25.23 | 0.93 | 0.05 | 30.15 | 0.97 | 0.05 | 34.05 | 0.98 | 0.02 | 29.45 | 0.96 | 0.08 |
| K-Planes | 25.48 | 0.95 | - | 31.79 | 0.98 | - | 33.72 | 0.98 | - | 32.64 | 0.98 | - |
| HexPlane | 25.22 | 0.94 | 0.04 | 30.67 | 0.98 | 0.03 | 34.36 | 0.98 | 0.02 | 31.65 | 0.97 | 0.04 |
| WIM | 13.95 | 0.72 | 0.28 | 19.05 | 0.87 | 0.14 | 16.26 | 0.89 | 0.12 | 17.95 | 0.87 | 0.16 |
| Ours | 25.08 | 0.92 | 0.07 | 32.24 | 0.98 | 0.03 | 34.46 | 0.98 | 0.02 | 32.22 | 0.98 | 0.03 |

the merging threshold discussed in our paper, shown as the rectangles in the figure. Note that the same color in different steps does not denote the same group.

### A.4 MOTION GROUPING EVALUATION DETAILS.

We utilized Kubric (Greff et al. (2022)), a data generation pipeline for multi-object videos with annotation, to create our evaluation dataset. The MOVi-C pipeline from Kubric was employed to generate scenes, utilizing realistic, textured objects from the GSO dataset (Downs et al. (2022)). To create a diverse set of scenes, a random HDRI was used to generate the background. The number of objects in each scene varied randomly between one to five, and object shadows were not considered due to their highly non-rigid nature. Objects were placed randomly in space with a randomized initial velocity. To capture each scene, the camera was positioned above the objects with a camera movement setting similar to Pumarola et al. (2020). Finally, each scene was simulated and rendered into a 120-frame image sequence, with ground truth instance masks. To evaluate the mean Intersection over Union (mIOU) between our grouping result and the ground truth mask, it is necessary to establish correspondence between them. To achieve this, we utilized the first 10 frames in the sequence to determine the correspondence, which then was extended to the whole sequence. For each group, we tallied the number of pixels belonging to each label in the first ten frames and identified the ground truth label with the highest number of pixels to associate it with the group. This operation enabled us to map each group to a label and convert the group map into a label map. Subsequently, we calculated the per-frame IOU between the converted label map and the ground truth mask. To obtain the final mIOU, we averaged the IOUs of all labels over the entire image sequence.

### A.5 TRAINING DETAILS AND HYPER-PARAMETER SETTINGS

We use $50 \times 50 \times 50$ voxels to construct the Eulerian volume $\mathbf{V}_E$ and the Lagrangian volume $\mathbf{V}_L$ with a feature dimension of 20. The canonical volume $\mathbf{V}_c$ is constructed with a $160 \times 160 \times 160$ voxel. Following Fang et al. (2022), the feature dimension of $\mathbf{V}_c$ is set as 6. To alleviate the optimization difficulty and speed up training, we set the initial resolution of the canonical volume to $40^3$ and upsample it at the $4k$, $6k$, and $8k$ iterations. For the MLPs in our framework, we set the channel number of all hidden layers to 128. We use two-layer MLPs for the motion extractors $\mathcal{E}_E$ and $\mathcal{E}_L$. The parameters of the rigid transformation decoders, $\mathcal{D}_E$ and $\mathcal{D}_L$, are shared. This sharing ensures that the decoded motion parameters exhibit consistency when the motion features are consistent. Two separated linear layers are used for the decoders to predict the 6D rotation and 3D translation with biases as $(1, 0, 0, 0, 1, 0)$ and $(0, 0, 0)$, respectively so that the initial deformation is an identity. For motion grouping, we set the slot number to 12 and initialize the slots from a standard normal distribution. We use the Adam optimizer for a total of $20k$ iterations, by sampling 4096 rays from a randomly sampled image in each iteration. To reduce the learning difficulty, we add images into the training set progressively at the early training stage. We set the learning rate as 0.08 for the Eulerian and Lagrangian volumes, 0.01 for the canonical volume, $6 \times 10^{-4}$ for $\mathcal{E}$ and $\mathcal{D}$, $8 \times 10^{-4}$ for other networks. For the overall loss function, we described it as:

$$\mathcal{L} = \mathcal{L}_{\text{photo}} + w_{\text{cycle}}\mathcal{L}_{\text{cycle}} + w_{\text{per\_pt}}\mathcal{L}_{\text{per\_pt}} + w_{\text{entropy}}\mathcal{L}_{\text{entropy}} + w_{\text{tv}}\mathcal{L}_{\text{tv}}. \tag{15}$$

We use $\mathcal{L}_{\text{per\_pt}}$ and $\mathcal{L}_{\text{entropy}}$ to supervise the color of sampled points and regularize the background probability, respectively. We set $w_{\text{per\_pt}}$ and $w_{\text{entropy}}$ to 0.01 and 0.001. To encourage reciprocity of motion between the Eulerian and Lagrangian views, we use the cycle consistency loss $\mathcal{L}_{\text{cycle}}$ with a weight parameter $w_{\text{cycle}}$ of 0.1. Additionally, we apply the total variation loss to smooth the features of the motion volumes $\mathbf{V}_E$ and $\mathbf{V}_L$:

$$\mathcal{L}_{\text{tv}} = \frac{1}{N} \sum (\sqrt{\triangle^2 \mathbf{V}_L} + w_E \sqrt{\triangle^2 \mathbf{V}_E}) \tag{16}$$

Where $N$ denotes the number of parameters of each motion volume, $\triangle^2$ represents the square difference between neighboring values in the motion volume, $w_E$ denotes the additional weight between $\mathbf{V}_E$ and $\mathbf{V}_L$. We set the $w_{\text{tv}} = 0.01$ and $w_E = 1$ for the D-NeRF synthetic dataset. For motion grouping evaluation, we decrease these two weights due to the more complex object geometry and motion patterns, we set the $w_{\text{tv}} = 0.001$ and $w_E = 0.1$.

### A.6 MORE RESULTS OF APPLICATIONS ON SYNTHETIC DATA.

Except for the real data application, we also conducted structural scene modeling and editing on robotic synthetic data. We utilized a subset of the robotic manipulation environments in ManiSkill (Mu et al. (2021)) to simulate robot-object interactions with a similar monocular setting as (Pumarola et al. (2020)). As shown in Figure 11, We conducted experiments using two different setups: articulated object manipulation and rigid-body manipulation. In the first setup, the robot performs operations on a particular movable part of an articulated object, as shown in Figure 11 above. In the second setup, the robot is tasked with grasping a specific object and moving it to a target position, as illustrated in Figure 11 below. In both setups, through observation, our method can accurately identify the manipulated object or object part, such as the drawer (orange) and the cabinet door (yellow), as well as the links and joints of the robot. What is more, benefit from the learned structural scene representation, we could apply direct high-level scene editing. In Figure 11, we show the editing operations like duplication, removal, scaling of a specific object in the scene, and the control of an articulated robot arm. Note that these operations are all performed in 3D space and rendered into 2D images for visualization.

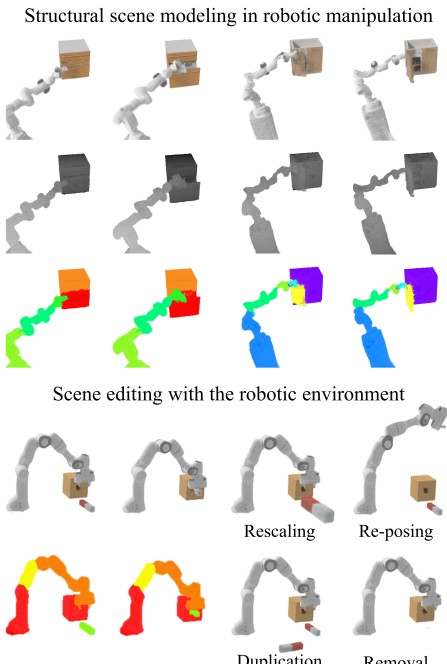

Figure 11: Structural scene modeling and editing in simulated robotic manipulation environments.

### A.7 IMPLEMENTATION DETAILS OF SCENE EDITING

Based on our part-level representation, we could directly edit the 3D dynamic scenes. In this paper, we present four scene-editing applications: Removal, Duplication, Rescaling, and Re-posing. Here, we will provide more details about how to implement these applications.

As shown in Figure 12, these application implementations could be divided into 2 ways. For Removal, after ray marching, the sampled points are fed into the Eulerian module to be transformed into canonical space. Then these canonical coordinates are grouped by our motion grouping network. To delete

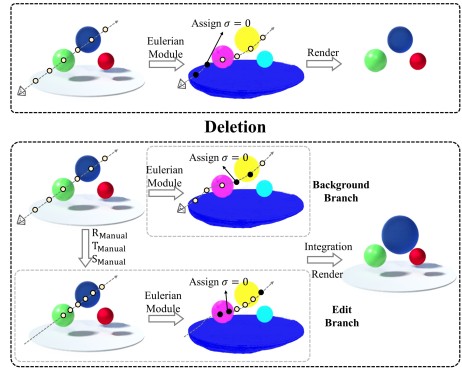

Figure 12: Illustration of the application implementation.

the specified group, we simply assign the density of the points in this group to zero to remove it from the canonical space. After rendering, the specific group will be deleted from the image.

Duplication, rescaling, and re-posing could be implemented in the same way. We do this by processing the group to be edited and the background (all the other groups) respectively. Specifically, the sampled points by ray marching are fed into the background branch and the edit branch. In the background branch, we leave the background group fixed and delete the group to be edited, which is the same as the Removal application. In the edit branch, we perform transformations for the target group, such as rotation, translation, scaling, etc. We achieve this by first performing a manually specified transformation on the sampled points. Then these transformed points are fed into the Eulerian module and the motion grouping network obtains the canonical coordinates and corresponding group index. After that, we set the density of the points in the background groups to 0 so that the transformation only acts on the target group. Finally, we integrate the corresponding points of the two branches to render them into the 2D image, the integration formula is as follows:

$$\sigma = \sigma_{bg} + \sigma_{edit}$$
$$\mathbf{c} = \frac{1}{\sigma}(\sigma_{bg} \cdot \mathbf{c}_{bg} + \sigma_{edit} \cdot \mathbf{c}_{edit}) \tag{17}$$

Note that by introducing additional edit branches, we could make different edits to different groups simultaneously.

