# OpenReview forum: "MovingParts: Motion-based 3D Part Discovery in Dynamic Radiance Field"
_ICLR.cc/2024/Conference — ICLR 2024 spotlight_

### Official Review · Reviewer_qQod · 2023-10-20

**Soundness:** 2 fair
**Presentation:** 2 fair
**Contribution:** 3 good
**Rating:** 8
**Confidence:** 5

**Summary:**

This submission takes monocular dynamic NeRF one steps further and tries to discover rigid objects parts. It uses D-NeRF-style monocular RGB input and aims for general articulated scenes. To this end, it uses forward and backward deformation modeling and a cycle consistency loss between them on the feature level (doing this on the features instead of offsets is novel). Then, at test time, the forward motions are merged into an adaptively determined number of rigid groups. Experiments evaluate the reconstruction and part segmentation quality. The results are almost exclusively on quasi-multi-view/D-NeRF-style synthetic scenes.

**Strengths:**

Extending dynamic NeRF towards part discovery is interesting, as also evidenced by a few related works in this direction.

The proposed method is quite straightforward and thus elegant. It does not appear to have overly involved components.

Using cycle consistency on motion features rathter than the final offsets is interesting and novel.

The reconstruction quality on the D-NeRF dataset is high (other datasets are unclear). The same goes for the part discovery, which gives very accurate looking qualitative results.

A few simple downstream applications like basic editing or tracking are presented.

**Weaknesses:**

Related Work and Method:

(A) I am not convinced that "Eulerian" and "Lagrangian" are the right terms. Standard deformable NeRFs like D-NeRF or Nerfies use backward deformation modeling. The paper proposes forward modeling instead. These terms should at the very least be mentioned explicitly somewhere. Both forward and backward modeling focus on particles and are direct functional inverses of each other. They thus resemble the Lagrangian perspective. As for example mentioned in Tretschk et al. State of the Art in Dense Monocular Non-Rigid 3D Reconstruction, I think it's more accurate to think of methods like NSFF that do not have a proper canonical model and rather directly regress the radiance field at each coordinate point in time and space as Eulerian. Such methods do not have a notion of particle, while both forward and backward deformation methods do. Or, to be more precise, they only use temporal consistency losses with auxiliary scene flow between *neighboring* frames and thus only have a notion of *instantaneous* flow to *temporally neighboring* frames at coordinate points in world space, which is exactly what Eulerian flow is. All of this is of course not ultimately relevant for the point of the paper in any way, only the presentation/writing, and I'm willing to hear why the notions proposed in the paper are more appropriate.

(B) The submission argues from the intractability of tracking a particle with backward deformation modeling. But for example Cai et al. Neural Surface Reconstruction of Dynamic Scenes with Monocular RGB-D Camera use an invertible deformation model that would allow to long-term track a particle in a straightforward manner. They lack expressivity and are computationally expensive but that's a more involved argument than the one in the submission.

(C) Liu et al. also propose cycle consistency of forward and backward modeling for dynamic NeRF in their NeurIPS 2022 paper DeVRF. It does this on the motion and not the feature level though. I think this should be discussed in the paper.

(D) It might be good to discuss how the proposed method relates to articulated NeRF methods like Wei et al. Self-supervised Neural Articulated Shape and Appearance Models.

(E) While not necessary, please consider citing Nerfies when introducing the idea of using an SE(3) parametrization, since they were the first to do that for dynamic NeRFs.

(F) Maybe mention that category-specific methods, e.g. those focusing on the human body or face, do quite easily allow for part understanding and editing/re-posing, etc. and that the proposed method focuses on general dynamic scenes.

(G) The lower half of page 5 talks about averaging all the features in a group. I assume all these features come from all the points in the batch? But what is in the batch? An entire image? I.e. how is good, reproducible coverage ensured such that averaging in this manner always gives pretty much the same final feature for a certain group across different views?

(H) What loss function provides a gradient for D_L, the Lagrangian decoder? All losses except for the cycle consistency loss only go into the canonical model and the Eulerian part. Where is the total variation loss applied exactly?

(I) I don't understand what the group merging module is used for. My only guess is this: During training, the initial, too-large number of groups is fixed/always used. Then, after training is finished, the group merging module is used to determine the final number of groups? So the APE cost is never used for backpropagation?

(J) The Lagrangian/forward module assumes articulated rather than fully non-rigid motion due to its per-group averaging operation. Please state this explicitly.

Experiments:

(K) I am concerned that video results are only shown for the synthetic D-NeRF dataset, which is effectively a multi-view dataset (as discussed in Gao et al. Monocular Dynamic View Synthesis: A Reality Check). I'd much prefer to see the video results of the real-world monocular scenes used in the paper. And the datasets from Figures 6 and 7 (although these also use the quasi-multi-view setting). Otherwise, I will assume that reconstruction only works for quasi-multi-view synthetic scenes and not for input that is truly monocular or real.

(L) Please add video results for part discovery on multiple scenes of the HyperNeRF dataset. Currently, only a handful of still images are shown. Otherwise, I will assume that these are severely cherry-picked and part discovery does not work on real-world scenes.

(M) The same applies for tracking. Using a well-textured canonical space like here https://github.com/pablopalafox/npms for example would help in visualizing the tracking/correspondence quality.

(N) Please add LPIPS scores everywhere. It's standard for NeRF papers (see e.g. the original NeRF paper) and PSNR and SSIM are quite similar, while LPIPS is much more perceptual.

(O) There appear to be no ablations. For example, applying the cycle consistency loss on features vs. positions. Or using SE(3) vs. offsets. And loss ablations, e.g.: Why is the per-point color loss useful? What happens without the total variation loss? Also, an ablation of whether the forward model helps *in reconstructing* would be good. Is this additional component useful? If it isn't, then the proposed part discovery portion of the method (i.e. the group merging module) might be added flexibly as a post-processing step to existing backward dynamic NeRFs to discover parts, which would be interesting to know.

(P) Can Watch-It-Move, the most relevant related work, not be applied to the D-NeRF dataset? I'm curious about the parts it would discover.

(Q) I assume that part segmentation is evaluated in 2D image space rather than in 3D due issues with the unobservable insides of objects?

**Questions:**

My questions are intertwined with the Weaknesses, please see above.

Overall, I unfortunately lean towards reject. I only have minor concerns about the method and related work, which can be easily addressed. However, the experimental evaluation is too lacking in multiple important regards. Still, if these concerns are addressed well in a rebuttal, I would most likely increase my score to accept.

---

> ### Author Response · Authors · 2023-11-17
> **Response to Reviewer qQod (Part 1)**
>
> Thank you for your detailed review and constructive suggestions. We will address your concerns as much as we can. We are also preparing an additional webpage for presenting the video results and ablations you mentioned in the experiments section. We will send it out soon.
>
> **(A) Concept of Eulerian and Lagrangian.**
>
> It is a good suggestion to explicitly mention forward and backward deformation in the paper and we will add it in a revision. We also agree with the reviewer that both forward and backward methods have some notion of particle. In particular, in D-NeRF or Nerfies, with the backward deformation, their volume density and color (the radiance field properties) are defined in the canonical space and can be thought of being models on the object material points/particles. However, please note that we focus on motion modeling (when it comes to Eulerian and Lagrangian views) here, and the backward deformation function itself is defined in the world coordinate space at fixed locations and cannot be considered as Lagrangian flow. In contrast, our forward deformation is defined in the canonical object coordinate space (on material particles) and exactly corresponds to what the standard Lagrangian flow is. To be more specific, our forward deformation can directly be used to describe the trajectories or pathlines of any object particles through the temporal frames, which is a classical lagrangian description; however, this is what the backward deformation in D-NeRF or Nerfies cannot achieve at all.
> Therefore, though the backward deformation is not identical to standard Eulerian flow (as the one the reviewer described in NSFF), we still consider the backward deformation as motions in an Eulerian description since they are defined at fixed world-space locations. We are happy to further clarify this in the paper.
>
> **(B) Intractability of tracking particles with backward deformation.**
>
> The reviewer is indeed correct that an invertible backward deformation model can achieve a similar goal because the forward deformation can be derived analytically. However, we think this actually aligns well with our argument that a forward deformation module is needed and an invertible function is an alternative way to achieve so. As pointed out by the reviewer, the invertible function also has other costs (lacking expressivity and computationally expensive) and our approach avoids these by modeling the two deformations separately with a cycle consistency loss. We will add the discussion with the invertible deformation in the paper.
>
> **(C, D, E, F) Discussions on DeVRF, Nerfies, Articulated NeRF, etc.**
>
> Thanks for pointing out the missing references. We will add and discuss them in a revision.
>
> **(G) Averaged group feature may be different across different views.**
>
> As you pointed out, the motion features utilized for grouping are sampled from points along rays in a single image, with 4096 rays sampled per batch. While there might be slight variations in the corresponding motion features across different views, we haven't observed any convergence issues.
>
> **(H) Gradient for Lagrangian module and TV loss.**
>
> The loss function for supervising the Lagrangian module includes the cycle consistency loss and total variation loss. We refer to this in the appendix.
>
> **(I) Group merging module.**
>
> Indeed, as you think, the group merge module is not engaged in training; it serves as a post-processing step to consolidate redundant groups.
> The APE cost is used as a quantitative measure of the disparities in motion patterns between groups, enabling the aggregation of similar groups.
>
> **(J) Piece-wise rigid motion.**
>
> Thanks for the suggestion. We primarily focus on general objects with piecewise rigid motion in this paper, and we will emphasize this in a revision.
>
> **(K, L, O, P) Video results, ablations and Watch-It-Move.**
>
> We are preparing an additional webpage for presenting the video results, ablations, etc. All these experiments' results will be shown in the next feedback soon.
>
> **(M) Tracking visualization.**
>
> Thank you for your suggestion. We will consider adding this kind of visualization to our paper.
>
> **(N) LPIPS.**
>
> Thanks for the comment, we will add the LPIPS score in a revision.
>
> **(Q) Part segmentation evaluation.**
>
> As you said, in section 5.3, we evaluated part segmentation on the 2D images. This is due to the fact that there is no guarantee that NeRF can model the geometry and motion of unobservable insides of objects.

---

> ### Author Response · Authors · 2023-11-21
> **Response to Reviewer qQod (Part 2)**
>
> Thanks again for your detailed review and valuable comments.
> We have revised and updated our draft based on the comments (highlighted in blue).
> In addition to this, we have created an anonymous website for providing relevant video visualizations, hope it can address your concerns.
> (https://anonymous418de.github.io/MovingParts/)
>
> In alignment with the order of your questions, below we provide our updates and feedback:
>
> **(A, J) Piece-wise rigid assumption and forward/backward deformation.**
>
> We emphasize our piece-wise rigid motion assumption and add the description of forward/backward deformation to align with existing dynamic NeRF methods in the Introduction section.
>
> **(C) Discussion on cyclic consistency loss of DeVRF.**
>
> In Section 4.4, we have introduced a discussion on the cyclic consistency loss of DeVRF. Additionally, in Section A.1, we have included an ablation study to compare their implementation with ours.
>
> **(D, F) Discussions on Articulated NeRF and Human NeRF.**
>
> We added references and discussions of articulated NeRF and human
> NeRF in the related work section.
>
> **(E) Discussions on SE(3) parametrization of Nerfies.**
>
> We added a discussion on Nerfies when using SE(3) as a motion parameterization, which adopts a similar representation to us (In Section 4.2).
>
> **(K, L) Video results.**
>
> We have visually represented all experimental results discussed in the paper, including the generated Kubric-based dataset and real-world data, etc, in the URL provided above.
> We hope these videos can help to address your concerns.
>
> **(N) LPIPS score.**
>
> In our revised version, we have incorporated LPIPS scores for all tabular data and discussions.
>
> **(O) Ablation Study.**
>
> In Appendix A.1, we have included ablations on the Lagrangian module, cycle consistency loss, and more. To demonstrate the visual difference of these ablated models, we have provided the corresponding video results of these ablation experiments on the provided webpage.
> As you mentioned, the presence of the Lagrangian module causes a slight drop in the rendering quality, which we consider to be due to the constraints of the rigid body parts of the Lagrangian module. But the quality drop is very marginal (less than 0.2dB); in general, our Lagrangian module allows for part discovery without losing the high rendering quality.
> It would be interesting to explore if a post-processing step with our module could also work for part discovery. But we think jointly training the two modules at the same time could likely better regularize the motion features, making them more meaningful for parts and leading to better part discovery.
>
> **(P) Watch-It-Move on D-NeRF dataset.**
>
> On the provided webpage, we have included visualizations of the Watch-It-Move results on the D-NeRF dataset. Notably, the monocular setup of D-NeRF presents a substantial challenge for Watch-It-Move due to its reliance on multi-view information for scene construction.
> Although it can approximate the images within the training views, it encounters a failure when applied to the test view.

---

> > ### Comment · Reviewer_qQod · 2023-11-21
> > **Thank you for the rebuttal!**
> >
> > I want to thank the authors for their rebuttal. I think it addresses the points raised by me and the other reviewers quite well. I find the ablation study in particular very well done and clarifying. It seems like the Lagrangian does not affect the reconstruction noticeably in either direction (good or bad), which is quite interesting to know. The only downside of the submission that is not really addressed is the small motion in the evaluation. However, monocular dynamic NeRFs have difficulties with large motion, which only works on the D-NeRF dataset because it's almost multi-view in its camera movement. Doing part discovery on top doesn't strike me as necessarily simplifying.
> >
> > The only question I'm still not clear on is (H). I look at Fig. 2: the computational graph starts in the top left (V_E), goes to the right (f_Em) towards the very right (x_C, V_C), goes down to (V_L) and then to the left (f_Lm) and then finally ends on the very left (x). As far as I can see, the cycle consistency works on f_Em and f_Lm. D_L is positioned *after* f_Em and f_Lm, so it shouldn't get any update from this loss. The total variation loss acts on V_E and V_L, which both also come before D_L in the computational graph and hence there shouldn't be any gradient into D_L either. It would be helpful if you could point out where in the appendix this is explained.
> >
> > I have a handful of small, unimportant comments:
> >
> > - On page 3, there is a blue sentence starting with lower-case "we".
> >
> > - In Table 2 in the appendix, please put "ours" behind "A1", to make it easier to parse the table at a glance.
> >
> > - It would be good if the clarification on (I) could somehow be squeezed in there. E.g. you could replace the first sentence in the paragraph with "Once training is finished and the weights are fixed, we further merge the groups to discover the final, minimal number of parts."
> >
> > - I am lost as to what "articulated objects (even more constrained than our rigid assumption)" in the rebuttal means. Aren't articulated and piecewise rigidity almost the same thing? Articulated to me means skeleton-based skinning, where the skeleton deforms in a piecewise rigid manner. I'm just wondering what is meant here by "more constrained". That the skeleton is assumed to be given as input? Because skinning also allows for smooth transitions between rigid parts, which the submission is too constrained to allow.
> >
> > Overall, I think this paper should be accepted.

---

> ### Author Response · Authors · 2023-11-22
> **Response to Reviewer qQod (Part 3)**
>
> We are very grateful for your response and support of our work. We are happy to see that our revision and additional clarification have addressed most of your concerns.
> In this revision, we have corrected the typos, and also changed the part you mentioned to help make the paper clearer for readers.
> Here are our responses to each of your feedback:
>
> **Gradient for optimizing D_L.**
>
> Thank you for pointing it out.
> It's worth noting that the rigid transformation decoders, D_L and D_E, actually share parameters.
> As a result, when employing cycle consistency loss to ensure consistent motion features across both views, it leads to their decoded rigid transformation parameters being consistently aligned as well.
> Therefore, the parameters of D_L and D_E (which are, in fact, the same network) could be optimized by the gradient in the Eulerian branch.
> In this revision, we have modified the corresponding part in Appendix A.5 to emphasize this point.
>
> **Typos and other modifications in paper.**
>
> Thank you for your careful observation, we have corrected this typo in the revised version.
> Also, we have put "Ours" behind "A1" in Table 2 to highlight our full model in these ablation studies.
> In Section 4.5, we have emphasized that our group merge module is a post-processing for adjusting the number of parts, which does not affect training.
>
> **Articulated objects.**
>
> We'd like to point out that articulated objects generally consist of multiple rigid components that are connected by joints. Although these components deform in a piecewise rigid manner, their motion is constrained by the joints, which limits the freedom of possible rigid motions. Specifically, 'Watch-it-move' explicitly models the joints of an articulated object in their method and is limited to handling a single such object. In contrast, our approach does not impose additional joint/connectivity constraints and instead assumes general piece-wise rigid motion. As shown in many examples in our paper, our method effectively handles scenes with multiple disconnected objects moving independently in space, extending beyond the scope of articulated object motion.

---

> > ### Comment · Reviewer_qQod · 2023-11-22
> > **Thank you again**
> >
> > Thank you for these clarifications!

---

### Official Review · Reviewer_NTzN · 2023-11-06

**Soundness:** 3 good
**Presentation:** 3 good
**Contribution:** 2 fair
**Rating:** 8
**Confidence:** 5

**Summary:**

In this paper is proposed a NeRF-based approach for simultaneous non-rigid scene reconstruction and rigid part grouping from monocular pictures. To this end, the method combines location-based Eulerian and particle-based Lagrangian interpretations, being a slot attention-based motion grouping network as part of the Lagrangian module. Both interpretations are a hybrid representation of volumetric features, which are supervised by a cycle-consistency loss. Every motion group is expressed by a rigid transformation per temporal frame. Experimental results are provided on synthetic and real datasets, obtaining a good trade-off between accuracy and computational cost in comparison with state-of-the-art techniques.

**Strengths:**

- The paper addresses an important problem in computer vision and learning. The proposed solution is a good combination of multidisciplinary areas.

- In general terms, the paper is well written and clear enough.

**Weaknesses:**

- The experimental evaluation is provided on simple synthetic datasets where the amount of deformation is a bit limited. Considering the paper formulation, I believe the method could handle strong deformations with no problem, but that is never validated.

- No full evaluation on real monocular image sequences.

**Questions:**

The motivation of the paper is good and the contributions are clearly stated. The technical section is well written and is clear enough. Most of the relevant details are provided as well as the explanations and discussions of every module the authors use.

The combination of Eulerian and Lagrangian formulations to handle the time-varying problem is a very interesting contribution. The authors exploit both points of view in a unified formulation.

In comparison with other NeRF algorithms to handle non-rigid objects, the algorithm in this paper can also infer a segmentation, exploiting the local rigidity of the clusters. This is a good point, but also shows us the type of deformation to be considered is a bit limited, as just piecewise rigid deformations are possible. In any case, the joint estimation is a good contribution. Similar to NeRF approaches, the camera poses are assumed in advance, representing, probably, a strong prior in this context.

The proposed method provides a good trade-off between accuracy and computational cost in comparison with other competing recent works. Moreover, the method can infer the part-based grouping automatically, as Noguchi et al. CVPR 2022 did. The results in terms of computational cost are really relevant.

Lack of realistic experimental evaluation. Unfortunately, the method is mainly tested on the D-NeRF synthetic dataset. I would like to see experimental results on real and challenging sequences. In addition to that, in spite of claiming non-rigid motions, the level of deformation on that dataset is very limited, and therefore I consider it is not the best one for evaluating non-rigid scenarios. In other words, the type of motion is piecewise rigid (as it was considered previously), that essentially is not the same as non-rigid. In fact, this can be observed in the estimation of groups. As it can be seen, the number of groups is very reduced and the global behavior in terms of deformations of those groups is quite simple. Again, the use of the dataset ManiSkill is not the best way to validate the strengths of the method due to the simplicity of the motion. In contrast, the authors could consider some real videos with dynamic scenarios to finish the full analysis and claim validation. I like real experiments in the context of scene editing.

The video is complete and definitely can help the reader.

---

> ### Author Response · Authors · 2023-11-17
> **Response to Reviewer NTzN (Part 1)**
>
> We are grateful to you for your time and the review. We will clarify some of your questions or concerns in this feedback.
>
> **Non-rigid scenarios.**
>
> Please note that the main objective of our method is to discover rigid parts in high-quality dynamic NeRF reconstruction.
> The rigid motion patterns are used as the evidence of part discovery.
> Therefore, we make explicit the assumption of our input here, which is the general object with piece-wise rigid motion.
> We emphasize that the automatic discovery of 3D rigid parts (with rigid motion) in dynamic NeRF is already a highly challenging task and it remains relatively unexplored in prior research.
> Watch-it-move is the (only) one previous work that has a similar goal to ours, but they can only handle articulated objects (even more constrained than our rigid assumption) and does not work with monocular input.
> We not only achieve automatic part discovery from monocular input but also achieve high rendering quality on par with previous dynamic NeRF methods that primarily focus on rendering quality and do not address part discovery at all.
> Please let us know if there are any additional experiments you would like us to undertake under the rigid motion assumption.

---

> ### Author Response · Authors · 2023-11-21
> **Response to Reviewer NTzN (Part 2)**
>
> Thank you once again for your insightful reviews.
>
> In this revision, we emphasize our piece-wise rigid motion assumption in the introduction section of the updated paper (highlighted in blue).
> It is crucial to note that even under this assumption, the automatic discovery of rigid parts from a dynamic NeRF remains a highly challenging task.
> As Watch-it-move is the (only) previous work that had a similar goal to ours, we visualized its results on our used D-NeRF dataset.
> The monocular setup poses a significant challenge to their method and leads to severe artifacts, as their method relies on multi-view information.
>
> To facilitate a clearer understanding, we have constructed an anonymous website showcasing the visualization results mentioned above and we hope you can find it useful.
>
> https://anonymous418de.github.io/MovingParts/

---

> ### Author Response · Authors · 2023-11-22
> **Response to Reviewer NTzN (Part 3)**
>
> Thank you once more for your valuable reviews.
> Please kindly let us know if our response and updated paper/website have addressed your concerns.
> We are happy to answer your remaining concerns and questions if you have any.

---

> > ### Comment · Reviewer_NTzN · 2023-11-22
> > **Regarding the rebuttal**
> >
> > Thank you very much for the clarifications and comments for all the points in the review. That was really a help to me. After considering the rest of the comments and answers, I think the paper could be accepted. Thanks again for the effort!

---

### Official Review · Reviewer_WRpS · 2023-11-08

**Soundness:** 4 excellent
**Presentation:** 3 good
**Contribution:** 3 good
**Rating:** 8
**Confidence:** 4

**Summary:**

This paper proposes a method for part-aware dynamic NeRF. While dynamic NeRF (D-NeRF) encodes the temporal information as well as the spatial radiance fields, it has not been easy to extract the object part information. To address this problem, this paper uses an idea similar to traditional motion segmentation, which combines the flow between adjacent frames (i.e., Eulerian flow in terms of particle flow simulation) as well as the clusters of motion trajectories (i.e., Lagrangian flow). Experiments show that the proposed method achieves near-SOTA accuracies for novel view synthesis with part-aware representation.

**Strengths:**

+ The proposed method correctly brings the idea of using motion trajectories to NeRF-based representations. In particular, the use of a cycle consistency loss that improves the agreement between Eulerian and Lagrangian motion shows a nice coupling of traditional ideas and modern neural networks.

+ The proposed method focuses on scenarios using a monocular moving camera, which makes the proposed method practical. In fact, the paper shows two practical examples of applications to robotics and scene editing.

**Weaknesses:**

- Using long-term motion trajectories is itself a traditional idea in computer vision, especially for motion segmentation (e.g. [a]). While I think that using Eulerian vs. Lagrangian views is a nice introduction to temporal motion representations, it would be nicer if they also briefly discussed the relationship of their idea to these traditional attempts. In my opinion, this would not sacrifice their technical contribution, but rather strengthen the theoretical and academic value of their approach.

[a] Keuper, Margret, Bjoern Andres, and Thomas Brox. "Motion trajectory segmentation via minimum cost multicuts." Proceedings of the IEEE International Conference on Computer Vision. 2015.

- Since their technical contribution may be to combine short- and long-term motion trajectories, readers may want to see the ablation studies comparing with and without these modules.

**Questions:**

Q. Related to the above, since the proposed method focuses on monocular input, readers may wonder if motion segmentation is applicable to their input sequence. In this regard, what happens if they apply motion segmentation to input sequences and then reconstruct *part-wise* NeRF?

---

> ### Author Response · Authors · 2023-11-17
> **Response to Reviewer WRpS (Part 1)**
>
> Thank you for the positive feedback, we address your problems below.
>
> **Traditional motion segmentation with long-term motion trajectories.**
>
> Thanks for pointing out the connection to motion trajectory and motion segmentation. We will cite the papers and add discussion in a revision. In general, different from the motion segmentation methods that mainly focus on recovering long-term point trajectories (spatio-temporal curves) in the 2D image plane from videos, our approach recovers 3D motion trajectories while ensuring multi-view consistency.
>
> **Part-wise NeRF with motion segmentation.**
>
> It is interesting to combine motion segmentation methods with dynamic NeRF approaches. This connection could enhance the establishment of long-term correspondences, thereby aiding in motion modeling and particle tracking for dynamic NeRF applications. However, it is challenging to generate consistent masks through the entire video, especially when there are strong occlusions, which might lead to blurry reconstruction or flickering rendering. Our approach instead naturally ensures consistency via joint optimization.
>
> **Ablations of the proposed modules.**
>
> Thank you for the suggestion, we are preparing an additional webpage for presenting more video results and ablations. It will be shown in the next response soon.

---

> ### Author Response · Authors · 2023-11-21
> **Response to Reviewer WRpS (Part 2)**
>
> Thank you once again for your insightful suggestions and valuable comments. Taking your feedback into consideration, we have implemented the following changes in the paper, which are highlighted in blue for your convenience:
>
> 1. We have incorporated a discussion on the traditional motion segmentation you mentioned. Please refer to the paragraph on Part Discovery from Motion in the related work section.
>
> 2. We have included ablation studies on our key modules and loss functions, and for detailed information, kindly consult Appendix A.1 in the revised version. Additionally, we have developed an anonymous webpage featuring video visualizations of the ablations and other experimental results.
> We believe it will provide a more intuitive understanding of the paper.
> (https://anonymous418de.github.io/MovingParts/)

---

> > ### Comment · Reviewer_WRpS · 2023-12-01
> > **After rebuttals**
> >
> > I thank the authors for their comments that clarify my concerns.
> > After reading the rebuttal and other reviews, I recommend that this paper be accepted as before.

---

### Official Review · Reviewer_5UoR · 2023-11-10

**Soundness:** 3 good
**Presentation:** 3 good
**Contribution:** 3 good
**Rating:** 8
**Confidence:** 3

**Summary:**

This paper presents an approach for dynamic scene reconstruction that allows the discovery of parts as a result of the factorization of the scene motion. The approach is based on monocular NeRF. A part corresponds to particles that share a common motion pattern. Scene motion is parameterized with both an Eulerian view and a Lagrangian view. The Lagrangian view allows tracking the particles on objects. Since the particles on a rigid part share a common rigid transformation pattern, a motion grouping module is used to detect the parts. The method is based on three modules, namely: the Eulerian module, the Lagrangian module and the canonical module. The Eulerian
module and the Lagrangian module observe the motion of specific spatial locations and specific particles, respectively. The canonical module serves to reconstruct the geometry and appearance for volume rendering. The the reconstruction and part discovery performance  of the method is experimentally evaluated on a synthetic dataset provided by D-NeRF. The metrics used for evaluation were PSNR and SSIM. The motion grouping is evaluated on a synthetic dataset created using Kubric toolkit. In both evaluations the method obtained performances better or at the level of the state-of-the-art.

**Strengths:**

The strength of the paper stems from being a method based on monocular NeRF that besides allowing to perform dynamic scene reconstruction, allows also the discovery of rigid parts. Another strength of the approach is that it used hybrid feature volume and neural network representation, which allows  fast convergence during training.

**Weaknesses:**

The grouping of the parts is based on predicted rigid motion. Rigid motion can be a significant constraint in many applications. The approach is inspired by fluid simulation, with the scene motion being observed from both the Eulerian view and Lagrangian view. The Eulerian and Lagrangian descriptions of a flow field are related by the material derivative. It is unclear whether the method ensures that such relationship is verified or met by the proposed approach.

**Questions:**

--Can you use other motion models, other than rigid ones?
--What is the relationship between the cycle consistency between the Eulerian and Lagrangian modules and the material derivative?
--How are the coordinates in the Eulerian and Lagrangian modules related?

**Details Of Ethics Concerns:**

Ethics is not relevant for the subject of this paper.

---

> ### Author Response · Authors · 2023-11-17
> **Response to Reviewer 5UoR (Part 1)**
>
> We sincerely thank you for your review and insightful thoughts. Below we would like to give detailed responses to your comments.
>
> **About the material derivative.**
>
> Our hybrid motion modeling is not directly connected to the material derivate. The material derivative describes the time rate of change of some physical property of a field, generally linking the Eulerian and Lagrangian descriptions of the same property's derivatives. While we adopt the Eulerian and Lagrangian perspectives from simulation, our hybrid model is designed to model the field motion/flow, instead of the time derivative of any additional field properties. We also do not define traditional field velocity in the model and hence there is no direct relationship with the material derivative. The goal of our designs is to build cycle consistency between the fixed (Eulerian) Cartesian grids and moving (Lagrangian) material points. It will be interesting to explore future extensions of our model to incorporate velocity and other material derivatives in dynamic fields.
>
> **The coordinates of the Eulerian and Lagrangian modules.**
>
> The Eulerian volume is defined with a fixed Cartesian grid in the world coordinate system and the Eulerian module maps each $x^w$ in the grid at any moment $t$ to the canonical space $x^c$ (i.e., the object coordinate system). In contrast, the Lagrangian volume is the same as the canonical space, aligned with the object at the specific moment $t_c$, and the Lagrangian module continuously tracks the trajectory of each object material particle $x^c$ from the canonical space to any moment $t$ according to the motion feature of each particle stored in the Lagrangian volume.
>
> **Motion models other than rigid ones.**
>
> In this paper, our goal is to achieve 3D rigid part discovery in dynamic NeRF reconstruction in an unsupervised manner. Please note that this is already a highly challenging task and relatively unexplored by prior arts. Watch-it-move is the (only) previous work that has a similar goal to ours, but they can only handle articulated objects (even more constrained than our rigid assumption) and does not work with monocular input. We not only achieve automatic part discovery from monocular input but also achieve high rendering quality on par with previous dynamic models that focus on rendering only. Potentially, our model can also be extended to handle more complex non-rigid motions in the future. However, the model might require other motion priors (e.g. motions of a specific kind of object), with ideally another low-dimensional motion parameter space, since we solve this problem in a per-scene optimization pipeline without additional part supervision.

---

> ### Author Response · Authors · 2023-11-21
> **Response to Reviewer 5UoR (Part 2)**
>
> Thanks again for the positive feedback.
> In response, we have created an anonymous webpage that offers an extensive collection of video results, Please check it out if you are interested in it.
>
> https://anonymous418de.github.io/MovingParts/

---

> > ### Comment · Reviewer_5UoR · 2023-11-23
> > **Thank you for your answers and additional results.**
> >
> > Your answer addressed my concerns. Furthermore, I also think that you addressed almost all the issues raised by the other reviewers. The video results provided in the anonymous webpage were helpful. I think that this is a good paper that should be accepted.

---

### Meta-Review · Area_Chair_zG9q · 2023-12-06

**Metareview:**

This paper presents a dynamic scene reconstruction that allows the discovery of parts as a result of the factorization of the scene motion. Based on monocular NeRF, the proposed method combines location-based Eulerian and particle-based Lagrangian interpretations, and they are supervised by a cycle-consistency loss. The reconstruction and part discovery performance of the proposed method is experimentally evaluated on the D-NeRF 360 synthetic dataset to demonstrate the effectiveness of the proposed method.  Extending dynamic NeRF to rigid part discovery is highly appreciated by the reviewers. The usage of a cycle consistency loss on motion features was also appreciated. The reviewers raised constructive concerns to improve the paper including more appropriate usage of terms such as piecewise rigid motion and the relation between forward/backward and Eulerian/Lagrangian.  Some missing papers were also pointed out in the constructive way.  The authors addressed the raised concerns properly in their rebuttal and followed discussion between the authors and the reviewers.  At the end of discussion, all the reviewers agree to accept the paper by acknowledging sufficient contributions of the paper.  The paper should be accepted, accordingly.

**Justification For Why Not Higher Score:**

The proposed dynamic scene reconstruction that allows the discovery of parts as a result of the factorization of the scene motion is innovative and novel enough to see a spotlight paper.  Although the proposed method is novel enough, it is not an oral level paper.

**Justification For Why Not Lower Score:**

The reviewers unanimously support this paper.

---

### Decision · Program_Chairs · 2024-01-16

Accept (spotlight)